# Stereotactic Salvage Radiotherapy for Macroscopic Prostate Bed Recurrence After Prostatectomy: STARR (NCT05455736): An Early Analysis from the STARR Trial

**DOI:** 10.3390/cancers17132092

**Published:** 2025-06-23

**Authors:** Niccolo’ Bertini, Giulio Francolini, Vanessa Di Cataldo, Pietro Garlatti, Michele Aquilano, Giulio Frosini, Olga Ruggieri, Laura Masi, Raffaela Doro, Mauro Loi, Pierluigi Bonomo, Daniela Greto, Isacco Desideri, Gabriele Simontacchi, Icro Meattini, Riccardo Campi, Lorenzo Masieri, Lorenzo Livi

**Affiliations:** 1Department of Biomedical, Experimental and Clinical Sciences “Mario Serio”, University of Florence, 50134 Florence, Italy; 2Radiation Oncology Unit, Azienda Ospedaliera Universitaria Careggi, University of Florence, 50134 Florence, Italy; 3CyberKnife Center, Istituto Fiorentino di Cura ed Assistenza, 50139 Florence, Italy; 4Unit of Urological Robotic Surgery and Renal Transplantation, Careggi Hospital, 50134 Florence, Italy; 5Department of Oncologic, Minimally-Invasive Urology and Andrology, Careggi Hospital, University of Florence, 50134 Florence, Italy

**Keywords:** radical prostatectomy, adjuvant therapy, prostate cancer

## Abstract

This study investigates the safety profile and biochemical response associated with stereotactic salvage radiotherapy (SSRT) administered for visible tumor recurrence in the prostate bed following radical prostatectomy. Conducted as part of a prospective, multi-institutional clinical trial, the research provides valuable insight into the effectiveness and tolerability of this targeted treatment approach. The findings suggest that SSRT is generally well tolerated by patients, with minimal severe side effects reported. Furthermore, the treatment appears to offer promising biochemical control of recurrent prostate cancer. These results support the potential of SSRT as a viable salvage therapy, emphasizing its role in improving clinical outcomes in a challenging patient population.

## 1. Introduction

Prostate cancer remains one of the most common malignancies in men, and radical prostatectomy (RP) is a well-established treatment for localized disease. However, up to 29% of patients experience biochemical recurrence (BR) within ten years of surgery [1]. For this reason, further therapeutic interventions are often necessary to manage recurrent disease and prevent its progression to metastatic stages. Salvage radiation therapy (SRT) after a radical prostatectomy offers a potentially curative treatment option in these cases and remains a base in the management of post-operative relapse [2]. Despite its demonstrated efficacy, patients with positive metabolic imaging indicating macroscopic recurrence often face poorer outcomes. This underscores the need for more personalized and intensified treatment strategies tailored to the biological and anatomical specifics of the recurrent tumor.

To address this clinical challenge, dose escalation in SRT has been proposed as a method to improve local disease control. Retrospective studies suggest that intensifying local treatment through higher radiation doses may lead to better clinical outcomes in patients experiencing macroscopic relapse confined to the prostate bed [3].

Moderately hypofractionated radiotherapy has been successfully tested for SRT [4]; however, strong prospective evidence is still needed in this scenario.

Nonetheless, a consensus on the optimal radiation dose, fractionation schedules, and techniques for such relapses remains elusive [5,6,7,8]. This is primarily due to variability in imaging modalities, patient characteristics, and institutional protocols. However, the advent of PSMA PET/CT and magnetic resonance imaging (MRI) has significantly improved detection and characterization of local recurrences, allowing for more precise treatment planning [9,10,11,12,13].

Moreover, the integration of advanced molecular imaging has shifted the paradigm from empirical to evidence-based radiation therapy. PSMA PET/CT, in particular, has demonstrated superior sensitivity and specificity in detecting even low-volume disease, thereby enabling early intervention before the spread of metastases. This capacity for early and accurate detection enhances patient selection for localized treatment and limits overtreatment with systemic therapies such as androgen deprivation therapy (ADT), which can negatively impact quality of life.

In this context, stereotactic salvage radiotherapy (SSRT) has emerged as a promising treatment modality. SSRT offers high-dose, highly conformal radiation delivery to a limited target volume in fewer treatment sessions compared to conventional SRT, potentially improving patient convenience and treatment compliance. Its ultra-hypofractionated approach may also have favorable radiobiological advantages, particularly in prostate cancer, which is known to have a low α/β ratio. Although still considered experimental, early studies and initial clinical experiences suggest that SSRT may offer superior disease control with potentially lower toxicity [14,15]. These promising outcomes have spurred further investigation into SSRT’s potential role in the treatment algorithm, including the STARR trial (STereotactic sAlvage Radiotherapy for macroscopic prostate bed Recurrence after prostatectomy—NCT05455736) [16]. This prospective trial was initiated to assess the safety and efficacy of SSRT in patients with macroscopic prostate bed recurrence and seeks to fill the current knowledge gap in the literature regarding optimal management strategies for this subset of patients.

This article presents an update of the first 51 patients enrolled in the STARR trial, focusing specifically on biochemical outcomes, radiological responses, and treatment-related acute and late toxicity, aiming to offer insights into both the efficacy and safety profile of SSRT in this clinical setting.

## 2. Materials and Methods

The STARR trial (NCT05455736) is a prospective, multicenter clinical study conducted at several leading oncological institutions in Italy. It includes patients previously treated with radical prostatectomy (RP) for localized prostate cancer who subsequently developed a macroscopic recurrence confined to the prostate bed. Recurrence was identified by a PSA level greater than 0.2 ng/mL and confirmed using advanced imaging techniques such as Choline or PSMA PET/CT. MRI was employed in all cases to confirm the recurrence unless contraindicated due to implanted devices or other patient-specific factors. Patients were excluded from the trial if they presented with regional or distant metastases, had any contraindication to SSRT such as inflammatory bowel disease, or suffered from severe urinary incontinence or urethral strictures following surgery. Moreover, patients with PSA persistence within 16 weeks after RP were also excluded to eliminate cases of residual rather than recurrent disease.

All patients underwent SSRT using the CyberKnife^®^ (Sunnyvale, CA, USA) robotic radiosurgery system, known for its ability to deliver highly accurate, hypofractionated radiation. Treatment consisted of 35 Gy delivered in five fractions administered every other day over approximately two weeks. For accurate target volume delineation, functional and anatomical imaging data from PET/CT and MRI were co-registered with planning CT scans using specialized software. The gross tumor volume (GTV) encompassed only the visible tumor in the prostate bed. A 2 mm margin was added to form the clinical target volume (CTV), and an additional 3 mm (reduced to 1 mm posteriorly to spare the rectum) defined the planning target volume (PTV). Fiducials were implanted in or near the recurrence site for real-time target tracking and alignment, and the InTempo™ system (Accuray Inc., Sunnyvale, CA, USA) adjusted imaging intervals based on tumor motion patterns, particularly in response to bladder and rectal filling variations.

Organs at risk (OARs) that were carefully contoured included the rectum, bladder, bowel loops, urethra, penile bulb, and femoral heads. These structures were prioritized during planning to minimize radiation-induced complications. No bladder catheterization or endorectal balloons were utilized, aiming for patient comfort and reproducibility of the bladder and rectal filling conditions. The treatment planning objective was to achieve optimal dose coverage of the PTV while respecting established dose constraints for all OARs.

All enrolled patients had discontinued androgen deprivation therapy (ADT) for at least 12 months prior to study inclusion, and the use of concurrent ADT during SSRT was not permitted. ADT was initiated based on PSMA detection of polymetastatic (more than 3 metastatic sites or visceral disease) disease at biochemical recurrence. Follow-up visits occurred at three-month intervals post-treatment and included detailed clinical evaluation, PSA testing, toxicity assessment made by treating physician, and imaging as clinically indicated. The first assessment of treatment response was conducted three months after SSRT completion, based on PSA measurements and clinical examination.

Complete biochemical response (CBR) and biochemical response (BR) were defined as a PSA nadir ≤ 0.2 ng/mL and ≤ 50% of baseline values, respectively [17,18]. Biochemical relapse was defined as a PSA rise above 0.2 ng/mL for patients achieving a PSA nadir after treatment ≤ 0.2 ng/mL, or a PSA rise > 25% of the nadir in two different measurements for patients with a PSA nadir > 0.2 ng/mL. Acute toxicities were recorded at each follow-up visit and graded using CTCAE version 4.03.

The primary endpoint of the STARR trial is the biochemical relapse-free survival (bPFS) rate at two years. Secondary endpoints include radiological progression-free survival (rPFS), ADT-free survival (aPFS), overall treatment-related toxicity, and quality-of-life measures. Kaplan–Meier curves were utilized for survival analysis, and descriptive statistics were used to summarize response rates, toxicity, and adverse events.

This analysis is focused on the first cohort of 51 patients enrolled.

## 3. Results

As of 17 November 2024, a total of 51 patients had been enrolled in the STARR trial. The demographic and clinical characteristics of the study cohort are outlined in Table 1. The average age was 71.0 years (range 58–82), with the majority of patients presenting with intermediate-risk prostate cancer at initial diagnosis. Approximately 38.4% of participants had a Gleason score of 3 + 4, while the remaining patients predominantly had Gleason 4 + 3 or higher. Regarding pathological staging, 57.7% of patients had pT2 or pT3a tumors, and 96.1% were node-negative (pN0) at the time of surgery. These baseline characteristics indicate a population with a generally favorable prognosis, albeit with localized relapse.

Most patients (n = 47) underwent restaging with PSMA PET/CT, which was performed at a median PSA of 1.34 ng/mL (range 0.3–4.5 ng/mL). Notably, 57.7% of patients had a PSA < 1 ng/mL at the time of imaging, underscoring the utility of early detection in identifying candidates for SSRT. The remainder of patients were staged with Choline PET/CT in centers where PSMA PET/CT was not yet available.

All 51 patients successfully completed the planned SSRT protocol with no treatment interruptions or early discontinuations. At the three-month follow-up, 23 patients (45.1%) achieved a complete biochemical response (CBR), while an additional 18 patients (35.3%) achieved a partial biochemical response (BR), resulting in a total biochemical response rate of 80.4%. These early biochemical responses suggest high initial efficacy of SSRT in controlling localized recurrence.

With a median follow-up period of 16 months (95% CI: 16–22 months), biochemical relapse was documented in 12 patients (23.5%). Among these, 9 patients exhibited biochemical progression without radiological evidence of disease, while 3 patients had documented radiological progression, all of which involved osseous metastases confirmed via PSMA PET/CT. ADT was initiated in six patients following disease progression, with the timing of hormonal therapy initiation determined by clinical judgment, PSA kinetics, and imaging findings.

At the time of this analysis, the median biochemical progression-free survival (bPFS), radiological progression-free survival (rPFS), and ADT-free survival (aPFS) had not yet been reached [Figure 1, Figure 2 and Figure 3], reflecting the relatively short follow-up duration and favorable early results. Kaplan–Meier analysis estimated the 1-year bPFS at 84.3%, rPFS at 94.1%, and aPFS at 88.2%. These outcomes demonstrate promising short-term disease control, particularly in a cohort that did not receive concurrent ADT.

Regarding toxicity, the treatment was well tolerated overall. Acute grade 2 gastrointestinal (GI) toxicity, primarily in the form of proctitis, was observed in two patients (3.9%). One patient (2.0%) experienced grade 2 genitourinary (GU) toxicity, manifested as dysuria. All acute toxicities were managed conservatively and resolved without the need for hospitalization or invasive interventions. There were no reported grade 3 or higher acute toxicities.

Late toxicity events were also infrequent. Only one patient experienced late grade 2 GI toxicity (chronic proctitis), which was successfully managed with medical therapy including topical corticosteroids and dietary modifications. No late GU toxicity greater than grade 1 was observed. These findings support the hypothesis that SSRT, when delivered with image-guided robotic platforms like CyberKnife^®^, can achieve high precision while maintaining a favorable safety profile.

## 4. Discussion

Overall, early results from the STARR trial suggest that SSRT in patients affected by prostate bed macroscopic relapse after RP is effective and well tolerated. Indeed, biochemical outcomes show that a significant percentage of patients had a rapid PSA decrease after treatment, with 76% of the population free from biochemical recurrence at the last follow-up. Management of macroscopic recurrence within the prostate bed is an important clinical need in the current clinical landscape. Advancements in imaging modalities, particularly the use of PSMA PET/CT, have enhanced the detection of macroscopic recurrences within the prostate bed, improving diagnostic capability and allowing for more precise targeting during salvage radiotherapy. This could lead to potential benefits in the postoperative salvage scenario [19,20].

Interestingly, all patients with biochemical recurrence after SSRT underwent PSMA PET/CT restaging, which offers significantly greater sensitivity than conventional imaging. [21].

Nonetheless, distant metastases after treatment were detected only in three patients, suggesting significant clinical benefit from SSRT in this population. Importantly, these outcomes were achieved without the use of concomitant androgen deprivation therapy (ADT), highlighting the potential of SSRT as a monotherapy in this selected population of patients with a good prognosis. Moreover, the use of baseline PSMA PET scan in a significant percentage of the population may have excluded the presence of subclinical distant or regional disease at follow-up, reducing the potential benefit of spatial cooperation between local treatment and ADT. Additionally, all postoperative disease persistences (defined as a positive PSA within 16 weeks from RP) were excluded, further improving patient selection in this cohort.

The role of ADT in conjunction with salvage radiotherapy remains a topic of ongoing research [21]. While some studies, such as the RADICALS-HD trial, have investigated the addition of ADT to postoperative RT, the benefits must be weighed against potential side effects. In the RADICALS-HD study, the addition of short-course ADT to postoperative RT did not improve clinical outcomes. Conversely, extending ADT to 24 months improved metastasis-free survival compared to 6 months of ADT, but this did not translate into an overall survival benefit after a median follow-up of nine years. Moreover, prolonged ADT is associated with adverse effects such as sexual dysfunction, metabolic syndrome, and osteoporosis, which can significantly impact patient quality of life [22].

The omission of ADT was intentional and reflects a central objective of our study: to evaluate the efficacy of stereotactic salvage radiotherapy (SSRT) alone in patients with macroscopic local recurrence. By excluding patients with microscopic or non-diagnostic disease, we focused exclusively on those with visible, PSMA PET-positive macroscopic relapse, allowing for a truly ablative local approach. The complete biochemical responses observed in our cohort are therefore attributable only to the local radiotherapy, without the confounding effect of systemic hormonal therapy. This highlights the potential of SSRT as an effective treatment modality in selected patients, while also preserving quality of life by avoiding ADT-related side effects.

The toxicity profile of SSRT was favorable, with acute GI and GU adverse events reported in two cases each, while only one G2 late GI toxicity event was detected. The ability to deliver escalated doses precisely to the macroscopic recurrence while sparing surrounding healthy tissues may contribute to these positive outcomes. In fact, all patients were treated with CyberKnife^®^, which has a steep dose gradient, thereby sparing the organs at risk. Moreover, robotic SSRT through the CyberKnife^®^ system could have further improved the tolerability profile. To further enhance this point, it is worth noting that a slight advantage of CyberKnife^®^ was also observed in the PACE B trial comparing ultrahypofractionated versus conventionally fractionated radiotherapy for definitive treatment of prostate cancer [23].

The SCIMITAR, SHORTER, POPART, and STARR trials collectively reflect the evolving landscape of post-prostatectomy radiation therapy, particularly focusing on the use of stereotactic body radiation therapy (SBRT). While they share a common goal—optimizing salvage radiotherapy—they differ in patient selection, technology, fractionation, and the use of androgen deprivation therapy (ADT), offering complementary insights into best practices [24,25,26].

SCIMITAR (Stereotactic MRI-guided Salvage Radiotherapy) is a prospective trial evaluating MRI-guided stereotactic body radiotherapy (SBRT) after prostatectomy. It aims to determine whether real-time MRI guidance reduces gastrointestinal (GI) toxicity compared to conventional CT-guided techniques. The trial highlights the advantages of adaptive planning and tighter margins made possible by superior soft-tissue visualization. Early results suggest a lower incidence of acute GI side effects due to improved accuracy and margin reduction. The SHORTER trial is a randomized phase II study comparing two MRI-guided salvage radiotherapy schedules: ultrahypofractionation (5 fractions) versus moderate hypofractionation (20 fractions). It assesses whether a shorter, more convenient regimen can be delivered safely without increasing toxicity or compromising patient-reported outcomes. The trial leverages MRI guidance for precise targeting, enabling dose escalation or fraction reduction with minimal side effects.

POPART is a prospective feasibility trial using CT-guided SBRT post-prostatectomy. Unlike SCIMITAR and SHORTER, it does not use MRI guidance but still reports low GI and GU toxicity, supporting the feasibility of postoperative ablative dosing with standard imaging, careful planning, and preparation.

A defining feature of SCIMITAR and SHORTER is the use of MRI-guided radiotherapy (MRgRT). In SCIMITAR, MRgRT significantly reduced acute gastrointestinal (GI) toxicity compared to CT-guided SBRT (41.9% vs. 72.5%), highlighting the benefit of enhanced soft tissue visualization and real-time tracking. SHORTER, as the only randomized trial among the group, directly compared ultrahypofractionated MRgRT (5 fractions) with moderately hypofractionated MRgRT (20 fractions), aiming to determine whether shorter courses can be safely delivered without compromising patient-reported urinary and GI quality of life. These two trials underscore the potential of MRgRT to safely intensify treatment while minimizing toxicity.

In contrast, the POPART trial used conventional CT-based SBRT with a non-MRI platform and still reported low toxicity rates, with no grade ≥ 2 late GU or GI adverse events. Although POPART had a smaller cohort and shorter follow-up, it supports the feasibility of delivering postoperative SBRT safely using standard image guidance, particularly when meticulous planning and patient preparation are applied.

Adding a different perspective, the STARR trial focuses specifically on patients with macroscopic recurrence within the prostate bed. All patients were treated with CyberKnife^®^, a robotic SBRT system offering steep dose gradients for enhanced precision. Impressively, 76% of patients remained free from biochemical recurrence at last follow-up without receiving ADT. However, due to the limited follow-up, comparison with existing literature remains difficult.

This sets STARR apart: it not only shows promising efficacy in a more challenging subgroup but also suggests that SSRT may be effective as monotherapy, particularly when precise imaging with PSMA PET/CT is used to guide treatment. Toxicity was comparable with the SCIMITAR and SHORTER trials, which use MRgRT, with only one case of grade 2 late GI toxicity reported.

The apparent benefits of MRgRT in this setting are likely due to the use of narrower PTV margins (3 mm instead of 5 mm with CTgRT). Notably, standard PTV margins for CTgRT with moderate hypofractionation typically extend up to 7 mm, making the 5 mm margins used in the SCIMITAR protocol already relatively narrow. The further reduction to 3 mm (1 mm in the posterior direction) in the STARR trial may effectively minimize rectal overdosing while also preventing underdosing of the clinical target volume.

While SCIMITAR and SHORTER explore the impact of MR guidance on toxicity and QOL, and POPART emphasizes broader feasibility, STARR contributes by addressing macroscopic disease and the role of SSRT without ADT.

A potential limitation of our study is the relatively short follow-up duration (16 months), which should be considered. Another limitation of the trial could be the currently incomplete case series and the single-arm design, which therefore does not provide any comparative data.

Future development involves completing the case series with 90 patients and assessing the outcomes with a longer follow-up. If the results are confirmed, a comparative study will then be proposed.

## 5. Conclusions

In summary, the interim results from the STARR trial suggest that stereotactic salvage radiotherapy (SSRT) is a promising treatment modality for patients with macroscopic prostate bed recurrence following a radical prostatectomy. The high biochemical response rate, low toxicity profile, and absence of concurrent ADT highlight the potential of SSRT as an effective monotherapy in well-selected patients. These findings support the integration of advanced imaging and precision radiotherapy in the salvage setting.

As the study progresses toward its target enrollment of 90 patients and longer follow-up data become available, it will be possible to more fully evaluate the long-term efficacy and safety of SSRT. If the current trends are sustained, SSRT could represent a promising option for this subset of prostate cancer patients, offering a high probability of disease control with minimal treatment burden.

Future research should aim to validate these findings in randomized controlled trials and explore the role of adjunctive therapies, such as short-course ADT or novel radiosensitizers, in enhancing outcomes [23,24,25,26,27,28,29,30,31,32]. Additionally, patient-reported quality-of-life data will be crucial in informing shared decision making and ensuring that therapeutic gains are achieved without compromising well-being.

Although the results are encouraging, they require confirmation through longer follow-up and support the development of prospective studies comparing stereotactic radiotherapy with conventional salvage treatments.

## Figures and Tables

**Figure 1 cancers-17-02092-f001:**
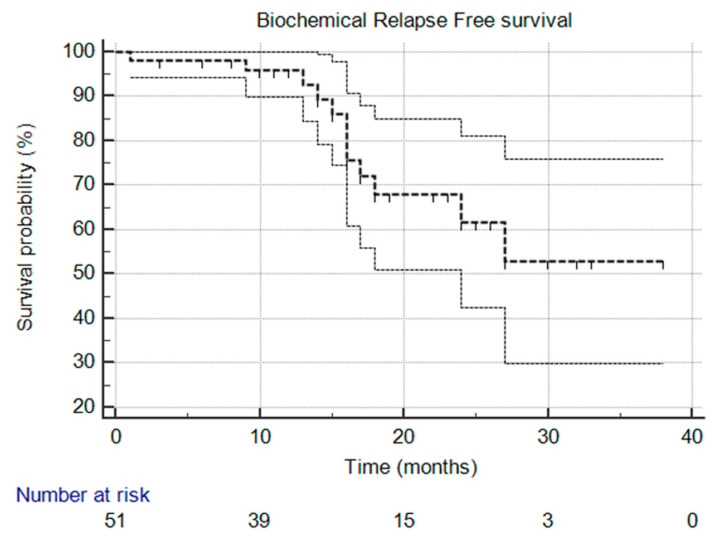
Kaplan–Meier survival curve that illustrates biochemical relapse-free survival over time. The median was not reached, and the 95% confidence interval is entirely above the 50% threshold, meaning the median is indeed not reached.

**Figure 2 cancers-17-02092-f002:**
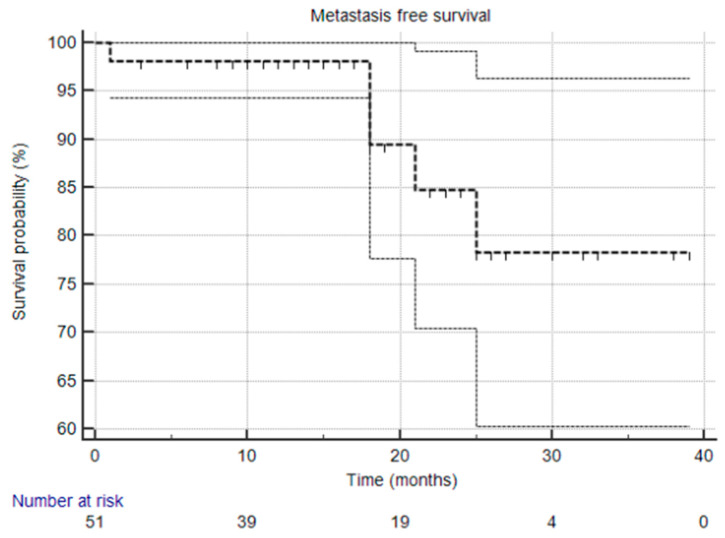
Kaplan–Meier survival curve that illustrates metastasis-free survival. The median was not reached, and the 95% confidence interval is entirely above the 50% threshold, meaning the median is indeed not reached.

**Figure 3 cancers-17-02092-f003:**
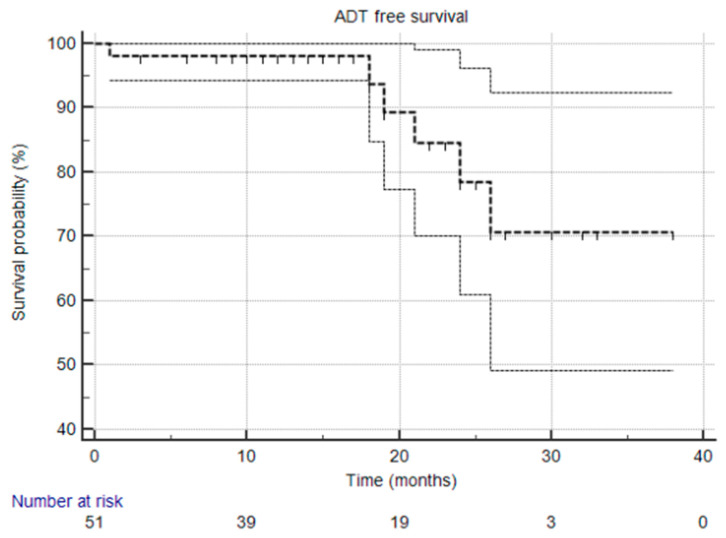
Kaplan–Meier survival curve illustrating ADT-free survival over time. The median was not reached, and the 95% confidence interval is entirely above the 50% threshold, meaning the median is indeed not reached.

**Table 1 cancers-17-02092-t001:** Principal baseline features of included patients.

	Value	Number of Patients and %	Number of Patients Without Informations and % (na)
**Average age of the patients**	71.0 years	51 (100%)	-
**Gleason Score**			
- 3 + 3	5	5 (9.80%)	-
- 3 + 4	19	19 (37.25%)	-
- 4 + 3	8	8 (15.69%)	-
- 4 + 4	9	9 (17.65%)	-
- 4 + 5	1	1 (1.96%)	-
- 5 + 4	1	1 (1.96%)	-
- na	-	-	8 (15.69%)
**pT stage (pT2, pT3a, pT3b)**			
- pT2	13	13 (25,49%)	-
- pT3a	17	17 (33.33%)	-
- pT3b	7	7 (13.73%)	-
- pT2c	6	6 (11.76%)	-
- na	-	-	8 (15.69%)
**pN stage (pN0, pN1, pNx)**			
- pN0	24	24(47.06%)	-
- pN1	2	2(3.92%)	-
- pNx	15	15(29.41%)	-
- na	-	-	10(19.61%)
**Risk**			
- R1	1	1(1.96%)	-
- R2	11	11(21.57%)	-
- R3	19	19(37.25%)	-
- na	-	-	20(39.22%)
**PSA at recurrence (PSArec)**			
- <0.5	17	17(33.33%)	-
- 0.5–1.0	12	12(23.53%)	-
- 1.1–2.0	9	9(17.65%)	-
- 2.1–3.0	5	5(9.80%)	-
- >3.0	7	7(13.73%)	-
- na	-	-	1(1.96%)
**PET PSMA (Yes/No)**			
- yes	41	41(80.39%)	-
- No	9	9(17.65%)	-
- na	-	-	1(1.96%)
**PET Coline (Yes/No)**			
- yes	7	7(13.73%)	-
- No	43	43(84.31%)	-
- na	-	-	1(1.96%)

## Data Availability

The data presented in this study are available on request from the corresponding author. The data are not publicly available due to privacy policy.

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
