# Peer review of "Stereotactic Salvage Radiotherapy for Macroscopic Prostate Bed Recurrence After Prostatectomy: STARR (NCT05455736): An Early Analysis from the STARR Trial"

_cancers, 2025, doi:10.3390/cancers17132092_

Round 1

Reviewer 1 Report

Comments and Suggestions for Authors

Manuscript Title: Stereotactic salvage radiotherapy for macroscopic prostate bed recurrence after prostatectomy: STARR (NCT05455736)

General Comments

The manuscript presents interim findings from the prospective STARR trial, investigating the safety and efficacy of stereotactic salvage radiotherapy (SSRT) using CyberKnife in patients with macroscopic local recurrence of prostate cancer following prostatectomy. The topic is timely and highly relevant, considering the increasing interest in dose-escalated, image-guided salvage radiotherapy. The authors demonstrate promising short-term biochemical control and low toxicity rates without concurrent ADT. However, the limited follow-up duration, lack of a comparator arm, and overinterpretation of interim results require cautious revision. Clarifying methodological elements, enhancing statistical reporting, and refining the narrative would significantly strengthen the manuscript.

Specific Comments

1. Abstract

The abstract effectively summarizes the study rationale, methods, and key results. However, phrases such as 'potentially effective cancer control' and 'urgent need' may overstate conclusions given the interim nature and limited follow-up.
Consider including explicit follow-up duration and sample size for context.
Minor grammatical edits would improve clarity.

2. Introduction

The introduction provides a strong clinical context and rationale for SSRT in macroscopic recurrence.
However, several references are cited repeatedly. Consider consolidating statements where possible.
The transition from historical management to the rationale for SSRT could be made more concise.

3. Materials and Methods

The methods are detailed and clinically sound. Patient selection and imaging protocols are well described.
Clarify how response definitions (CBR vs BR) were derived, and provide justification for chosen thresholds.
The criteria for ADT initiation after SSRT should be explicitly mentioned.
Mention whether toxicity was assessed by independent reviewers or by treating physicians.

4. Results

The results are encouraging, with strong response rates and minimal toxicity.
Include more detailed statistical measures (e.g., confidence intervals, p-values where applicable).
Kaplan-Meier curves are appropriate, but should be better annotated (e.g., events at risk, curve interpretation).

5. Discussion

The discussion effectively compares the STARR trial with SCIMITAR, SHORTER, and POPART trials.
 However, it is overly long and at times repetitive. Consider streamlining to focus on interpretation, limitations, and future directions.
Avoid overstating findings with terms like 'new standard of care' when based on interim data.
The impact of no ADT use deserves further emphasis as a distinguishing aspect of this cohort.

6. Conclusion

The conclusion is consistent with the presented data but should be more cautious due to the study’s interim nature and absence of a control arm.
Emphasize that further validation in a randomized controlled trial is essential before generalizing findings.

Conclusion

The manuscript requires several improvements to enhance clarity and strengthen scientific reporting. Please find below the key shortcomings identified:

1. The follow-up duration is short (median 16 months), limiting long-term conclusions on disease control, recurrence, or toxicity.

2. The single-arm design without a comparator arm prevents definitive conclusions about treatment superiority or equivalence.

3. Statistical analyses are basic and could be strengthened with multivariate or subgroup assessments, particularly to assess predictors of biochemical progression.

4. The language at times overstates the results (e.g., 'new standard of care') despite limited follow-up and small sample size.

5. The Discussion is lengthy and at times redundant—several paragraphs can be condensed to improve clarity and focus.

6. Some methodological details, such as exact criteria for initiating ADT upon progression and the basis for response classification (CBR vs BR), need clearer definitions.

7. Figures and survival curves (Kaplan-Meier) should be better annotated to enhance readability and interpretation.

Comments on the Quality of English Language

The manuscript is generally readable and scientifically sound but would benefit from professional English editing. Several sentences are lengthy and could be restructured for clarity and flow. Transitional phrases between paragraphs can be improved to enhance coherence. Some terminology is inconsistently applied (e.g., BR vs. CBR). Attention to grammar and conciseness would improve the overall presentation.

Examples of sentences needing revision:

  1. "Interestingly, all patients with a biochemical recurrence after SSRT underwent PSMA PET CT re-staging, a method with significantly higher sensibility if compared to conventional imaging."
    → Suggested: "Interestingly, all patients with biochemical recurrence after SSRT underwent PSMA PET/CT restaging, which offers significantly greater sensitivity than conventional imaging."

  2. "Impressively, 76% of patients remained free from biochemical recurrence at last follow-up, despite not receiving ADT, although it is a very interesting finding given the limited available follow-up, it is difficult to compare with other data from the literature, which are currently lacking."
    → Suggested: "Impressively, 76% of patients remained free from biochemical recurrence at last follow-up without receiving ADT. However, due to the limited follow-up, comparison with existing literature remains difficult."

Reviewer 2 Report

Comments and Suggestions for Authors

The paper evaluated the efficacy and safety of SSRT in patients with macroscopic prostate bed recurrence. Results indicated that SSRT is a safe and potentially effective option for macroscopic local recurrence, with encouraging response rates and minimal toxicity.

  1. The horizontal axis of the Kaplan-Meier survival curve is not marked with units, weeks or months.
  2. Kaplan-Meier survival analysis can be more detailed. The authors analyzed the response of different patient groups to treatment.

Reviewer 3 Report

Comments and Suggestions for Authors

The paper reports the first results of the STARR trial across Italian institutions on a small cohort of 51 prostate cancer patients that underwent stereotactic salvage radiotherapy for macroscopic prostate bed recurrence after prostatectomy. The results show that CyberKnife-based SSRT is feasible in this patient group leading to good tumour response and normal tissue tolerance. The study limitations are adequately identified and the future prospects mentioned as next steps will offer a larger picture on the efficacy of this radiotherapy approach on the select group of patients.

The paper is generally well written and clear. I only have a few comments to further improve the clarity / readability:

  1. The title should reflect the early nature of the trial results (and/or the small cohort) (‘an early updated analysis… / ‘interim results from the STARR trial’ or something similar.
  2. Please explain / justify the choice of 35 Gy delivered in 5 fractions against other hypofractionated radiotherapy schedules.
  3. In the Discussion section please add some comments on the possible dose-response relationship among patients with biochemical recurrence (dependence on the prostate bed volume or other factors).
  4. A subset analysis on those patients that presented biochemical recurrence might justify an adjustment on the dosage / fractionation schedule (or PTV margins) in high-risk patients.

Round 2

Reviewer 1 Report

Comments and Suggestions for Authors

I appreciate the authors’ comprehensive efforts in revising the manuscript titled “Stereotactic salvage radiotherapy for macroscopic prostate bed recurrence after prostatectomy: STARR (NCT05455736)”. The revisions have clearly addressed the key concerns raised during the review, including clarification of methodology, detailed justification for the patient cohort, and comparison with recent trials. The inclusion of updated data and expanded discussion enhances the scientific value and clinical relevance of the study. Thank you for your thoughtful and constructive responses.